# Divergent Selection Task Offloading Strategy for Connected Vehicles Based on Incentive Mechanism

**Senyu Yu** [1], **Yan Guo** [1,*], **Ning Li** [1], **Duan Xue** [1,2] and **Hao Yuan** [1]

1. School of Communication Engineering, Army Engineering University of PLA, Nanjing 210000, China; reborn63587742@163.com (S.Y.); js_ningli@sina.com (N.L.); 18635943760@163.com (D.X.)
2. School of Computer Science, Liupanshui Normal University, Liupanshui 553000, China
* Correspondence: guoyan_1029@sina.com

**Abstract:** With the improvements in the intelligent level of connected vehicles (CVs), travelers can enjoy services such as self-driving, self-parking and audiovisual entertainment inside the vehicle, which place extremely high demands on the computing power of onboard systems (OBSs). However, the arithmetic power of a single CV often cannot meet the diverse service demands of the in-vehicle system. As a new computing paradigm, task offloading based on vehicular edge computing has significant advantages in remedying the shortcomings of single-CV computing power and balancing the allocation of computing resources. This paper studied the computational task offloading of high-speed connected vehicles without the help of roadside edge servers in certain geographic areas. User vehicles (UVs) with insufficient computing power offload some of their computational tasks to nearby CVs with abundant resources. We explored the high-speed driving model and task classification model of CVs to refine the task offloading process. Additionally, inspired by game theory, we designed a divergent selection task offloading strategy based on an incentive mechanism (DSIM), in which we balanced the interests of both the user vehicle and service vehicles. CVs that contribute resources are rewarded to motivate more CVs to join. A DSIM algorithm based on a divergent greedy algorithm was introduced to maximize the total rewards of all volunteer vehicles while respecting the will of both the user vehicle and service vehicles. The experimental simulation results showed that, compared with several existing studies, our approach can always obtain the highest reward for service vehicles and lowest latency for user vehicles.

**Keywords:** intelligent transportation system; connected vehicles; vehicular edge computing; computational task offloading

## 1. Introduction

As the future direction of the transport industry, an intelligent transportation system (ITS) has already been developed in first-tier cities, and many other cities are also stepping up the construction of the relevant transportation infrastructure [1–3]. An ITS can not only reduce manpower and material resources, leading to better scientific traffic management, but can also analyze the road conditions in real time, improve traffic safety, and even save energy and reduce emissions. The above advantages of ITS are reflected and guaranteed by relevant technologies (cellular communication, cloud computing and edge computing), on-board applications (such as emergency warning, cooperative adaptive cruise [4], collision avoidance, and entertainment and multimedia applications) and supporting roadside infrastructure. However, with the increase in connected vehicles (CVs) and related equipment, a huge number of tasks or data are generated, and whether those tasks are processed in a timely manner is key to the normal operation of ITSs. The subsequent computing tasks should be offloaded to other terminals due to time limitations when the computing capacity of the current CVs cannot meet the requirements of the onboard system. One of the widely used computing methods, cloud computing, can offload data from CVs

or other intelligent terminals with insufficient computing power to remote cloud service centers and provide an auxiliary computing service. However, cloud computing also has its shortcomings. The main reason for this is that cloud service centers are generally far away from internet-connected terminals, leading to heavy network or propagation delays. At the same time, the number of terminals the cloud server can serve is also limited, while the number of intelligent and connected vehicles is rapidly increasing, meaning that the fixed remote cloud server is likely to be unable to meet the dynamic growth demand of terminal vehicle users. Mobile edge computing (MEC), a new technical way of balancing computing load, has entered the field of vision of many researchers to protect user privacy [5,6], improve efficiency and reduce the cloud computational load. Papers related to mobile edge computing have increasingly emerged in recent years. Mohammad Yahya Akhlaqi et al. studied over 200 of them and conducted a detailed analysis of the research fields, optimization objectives, algorithm design, evaluation indicators, data sets and tools used. They also presented visual qualitative or quantitative analysis results and finally proposed potential future research fields related to edge computing [7].

Various computing tasks and data generated by CVs or other smart devices could be offloaded to network edge devices with redundant computing capabilities, and this can be called computational offloading. Tasks can come from CVs with insufficient computing power, as well as from users' mobile devices [8], IoT devices [9,10] and devices from other industries such as healthcare enterprises (HEs) [11]. Those offloaded tasks or data are quickly processed, and the results are returned to meet the demands of different vehicular applications. Generally speaking, the terminal point of computational offloading of CVs is the roadside unit (RSU), the edge server, or the connected nearby vehicles with high computing power and redundant resources. CVs with lower computing power can offload computational tasks that cannot be completed locally to an edge server within the communication range or nearby intelligent vehicles for auxiliary computing to ensure the vehicle driver has a good user experience.

However, due to the special mobility of vehicles, CVs' computational offloading is different from that of IoT devices with relatively fixed locations. The topology of its communication network changes over time, and the resource retention of each CV is differentiated, which makes it complicated to reasonably select the offloading destination. There is also a contradiction between the intentions of the user vehicle (UV) and service vehicles (SV); that is, UVs with deficient resources or computational tasks to be offloaded expect nearby SVs to assist in their computation, while SVs with rich resources have no obligation to provide such services for UVs. This conflict hinders UVs' computational offloading, and intuitively means that UVs struggle to obtain a reliable quality of service (QoS).

In response to the above problems, this paper proposes a divergent selection computational offloading method based on the incentive mechanism (DSIM). Specifically, the main innovation points are as follows:

Firstly, considering that the infrastructure in some regions is incomplete or temporarily unavailable, and that there may be some available CVs with redundant resources near the UV, we no longer rely on the RSU or edge servers as the core processing equipment for computational offloading, but organize these nearby CVs into temporary opportunistic edge servers (resource pools). The voluntary SVs in this dynamic, self-organized temporary resource pool cooperate to provide services for UVs.

Secondly, we focused on the freeway scene that is currently lacking research. In this scenario, CV speed is faster, and the network topology changes more frequently. By modeling the mobile attributes of vehicles in the expressway, SVs in the resource pool are purposefully scored and comprehensively ranked. The computational tasks of the UV to be offloaded are classified according to its characteristics, and the offloading process is further refined so that the UV can choose the offloading destination preferentially.

Thirdly, to encourage available CVs in the vicinity to share resources, a comprehensive divergent selection incentive mechanism (DSIM) is proposed. When an available CV

becomes a member of the resource pool, it is rewarded according to its available communication time, computing power and service delivery time. The award is related to the unique ID of the vehicle and can be converted into virtual currency. In combination with the second point, both the UV and volunteer vehicles in the resource pool (also called SVs) have the right to choose independently. This divergent selection mechanism respects the wishes of both parties (UV and SVs) while maximizing the service quality of user vehicles.

The rest of this paper is organized as follows. The related work is outlined in Section 2. In Section 3, the system model and proposed divergent selection task offloading strategy based on the incentive mechanism (DSIM) is described in detail. The evaluation of DSIM method is presented in Section 4. We discuss our work in Section 5. The conclusion and future work are shown in Section 6.

## 2. Related Work

In 2016, Xueshi Hou et al. prospectively proposed using more and more internet-connected vehicles as servers to provide services for mobile users (vehicles, smart phones, smart wearable devices, etc.) [12]. The rational and efficient use of resources in connected vehicles has attracted the attention of many scholars. In particular, in 2019, 5G was put into commercial use in China, which made the interconnection between entities in the intelligent transportation system more convenient and efficient due to the high data rate (HDR) and low latency and flexible spectrum allocation, and also provided basic communication support for edge computing, computational offloading and other technologies. In recent years, with the further popularization of intelligent and connected vehicles and the optimization of on-board systems, a lot of CVs retained rich computing resources, and many scholars made use of them to provide services for other users (smart devices). Generally speaking, CV status can be divided into two categories. One is that, when the vehicle is parked at the roadside or parking lot, it is considered to be in a static state; the second is that, when the vehicle is driving on the road, it is in a moving state. Almost all the literature on vehicular computational offloading can be divided into one or both of these two categories. When CVs are stationary, their geographic location is completely fixed. For example, in the parking lot, the network topology of vehicle-to-vehicle (V2V) communication can be completely determined when no CV joins or leaves, which guarantees reliable computational offloading services. When CVs are in the mobile state, real-time changing network topology leads to a short communication time and dynamic resource distribution, which makes computational offloading in this scenario more complex. Static CV-assisted vehicular computational offloading is first introduced, and then we present the more convoluted mobile CV-assisted computational offloading, which is also the main focus of this paper.

### 2.1. Static CV-Assisted Task Offloading

Computing resources in stationary CVs need to be managed and integrated before they are effectively utilized, because they are randomly scattered in parking lots or highways. Some scholars propose establishing a fixed edge server at an appropriate location to manage these resources [13]. Specifically, the fixed server can collect all CVs' information within its communication range in real time. The computing tasks generated by mobile users are sent to the fixed edge server. The server processes these tasks first, but the upcoming tasks will be offloaded to stationary CVs with redundant resources for the second time, when there are too many tasks, or the server is overloaded. This method not only reduces the server load, but also meets a user's delay requirements. Similarly, on this basis, Yuwei Li et al. proposed a contract incentive mechanism based on game theory [14], where edge servers purchase resources from stationary CVs, and proposed that some contract treaties achieve information symmetry between SVs and UVs, thus encouraging CVs to contribute their resources and provide services for UVs. This significantly improves server utilization and reduces energy consumption.

In addition to CVs parked in the parking lot, many vehicles temporarily park on the roadside, which may also have a large number of available redundant resources. According to the research, the utilization rate of roadside parking spaces in urban roads has reached 93% [15]. Stationary vehicles in adjacent areas are integrated into clusters [16] and managed into temporary available servers. Then, temporary servers in different areas cooperate to cover a larger area and provide services for users in this range.

It can be seen that studies on edge computing assisted by stationary vehicles tend to focus on the edge server of fixed facilities to carry out unified management and resource allocation for parked vehicles. It is feasible and universal to build such edge servers and other infrastructure in large parking lots with fixed locations or densely populated urban areas. As mentioned above, the geographical location of the vehicle in motion is uncertain. It can be driven in city centers with a relatively complete infrastructure but can also be located in suburbs or remote areas where the infrastructure is not as perfect. The following provides a brief discussion of vehicular computational offloading research in this scenario.

### 2.2. Mobile CV-Assisted Task Offloading

Compared with CVs in a static state, vehicles in a mobile state change their position at all times, meaning that the network topology changes more frequently. In the meantime, the diversity of the resource capacity and the uncertainty of the communication time makes it more difficult for mobile-CV-assisted computational task offloading. Chen Chen et al. [17] of Xidian University took the lead in focusing on the computational task-allocation strategy on the highway in 2021. They used CVs near a UV as the resource pool (RP). A large computational task was divided into pieces by RSU and allocated to CVs in the resource pool according to their computing capacity. Finally, the deep reinforcement learning (DRL) method [18,19] was used to reasonably schedule the existing resources, and the total task completion delay was significantly reduced. However, this did not end dependence on roadside edge servers, acquiescing sound transportation's fundamental facilities, which is unrealistic.

Given that the computational resources of roadside static servers might be insufficient, driving CVs are integrated into clusters as a dynamic edge server [20]. The temporary dynamic server that is thus formed cooperates with the static server in the RSU and remote cloud server to provide computing services for user equipment (vehicles with insufficient computing power, roadside cameras and other devices). Among them, the static server in an RSU, as the core of the system, is responsible for processing the arrival requests of users and allocating them according to the existing resource distribution in order to meet the strict delay constraints of different tasks. Mobile and stationary CVs can be used as edge servers at the same time [21]. The computing tasks generated by users are first collected by roadside units, and then allocated to one of the base stations, mobile CVs and stationary CVs for computational processing according to the existing backup resources. Although the above papers consider the problem of the lack of computational resources of roadside servers, they still have not completely removed the core position of the RSU in vehicular computational offloading.

Digital twin (DT) and artificial intelligence (AI) technology can be combined to solve the complex situation of unbalanced computing power, unpredictable resource demands and dynamic network topologies [22], making full use of the resources of mobile CVs and minimizing the cost of computational task offloading. In addition to the network topology changing when CVs are moving, CVs' speed can also influence the task offloading process. That is, vehicles' speed and task type are linked, and their relationship and how this connection impacts the task offloading process can be discovered. The authors of [23] proposed an association model between CVs' speed and task type to make the task offloading decision more refined and targeted. However, the above studies did not factor in the willingness of available nearby CVs, which means that CVs provide services to a UV with default accessories but without complaint, which is, obviously, not fair. Even if nearby

CVs have redundant resources, they are not obligated to provide services to UVs, so certain incentives are needed to encourage these vehicles to provide services to users.

However, the following problems still exist: First of all, whether static or mobile-vehicle-assisted computational offloading is being used, the previous research takes RSUs or other servers with a fixed physical location as the core infrastructure to handle tasks or balance resources. However, at present, many countries still do not have a sound construction of intelligent transportation infrastructure. Remote areas or underdeveloped cities do not have reliable roadside units or edge servers as support, which causes serious obstacles to the implementation of the above research. Second, most of the existing research focuses on resource sharing or service improvements among intelligent vehicles in urban centers, where the available redundant resources are relatively sufficient due to their high density and the slow driving speed of urban vehicles. In contrast, there are also a huge number of CVs on freeways or suburban expressways. Meeting the computational offloading requirements of these rapidly moving UVs is extremely difficult and must urgently be solved when the number of vehicles within UV's communication range is small or the edge server is unavailable. Third, in order to improve the user experience (UE), UVs always tend to choose SVs with high computing power or efficiency to provide services. Meanwhile, nearby CVs with redundant resources are under no obligation to deliver a service for UVs [24]. The service can be considered as a "voluntary service". By default, no legal person must provide services for UVs without return for all connected vehicles with redundant resources, meaning that the service quality cannot be guaranteed.

Therefore, in order to supplement the shortcomings of the existing research, our proposed mechanism solves the computational offloading problem without relying on roadside edge servers. We analyze the motion patterns of vehicles, classify computational tasks based on priority, and respect the wishes of UVs and SVs. After finishing computational tasks, volunteer vehicles are rewarded. These reward values can be converted into virtual currency and used to deduct parking fees or highway tolls to meet the core needs of intelligent transportation systems—intelligence, convenience and efficiency.

## 3. Materials and Methods

Here, the system model for the task offloading process of high-speed connected vehicles is designed. The divergent selection mechanism is explained in detail in this chapter. First, the overall system model, high-speed traffic flow model, V2V communication model and computing model are introduced.

### 3.1. System Model

This paper focuses on vehicular computational task offloading on highways. The system model is shown in Figure 1. In fact, vehicles on the expressway have a higher speed and faster-changing communication topology than those on urban roads. Meanwhile, there are vehicles driving onto or out of the highway at intersections, making the choice of computational task offloading strategy more complex and ever-changing. In addition, many expressways are located at the edge of cities and rural areas, and the roadside infrastructure (such as roadside units (RSUs)) is not perfect, leading to unreliable computational offloading services for UVs (vehicle in LightSalmon in Figure 1). The vehicles in blue are non-volunteer CVs and are meaningless in the computational offloading process. Therefore, we studied how to make volunteer CVs self-organize into edge clouds to provide computation services for UVs when RSUs are unavailable. Volunteer CVs with redundant resources, SVs (vehicles in green in Figure 1), form a mobile opportunistic dynamic edge cloud or resource pool (cloud in green in Figure 1). Computation services are provided by the dynamic RP when UV requests arrive. The mobile opportunistic dynamic RP uses its existing resources in SVs to execute computing tasks and finally returns results. SVs that voluntarily provide their own resources are rewarded for their contribution. In this paper, focusing on the freeway scene where the roadside service unit is imperfect, the high-speed traffic flow model, the vehicle–vehicle (V2V) communication model and the computing task offloading model are

built. At the same time, vehicles with redundant resources in the traffic flow are afforded the right to choose whether to provide services to UVs. However, since these vehicles are not obligated to do so, we designed an incentive mechanism to encourage these vehicles to become voluntary vehicles in order to encourage owners of these vehicles to contribute their own computing resources. Accordingly, UVs can also select the destination for their computation offloading to obtain a better user experience.

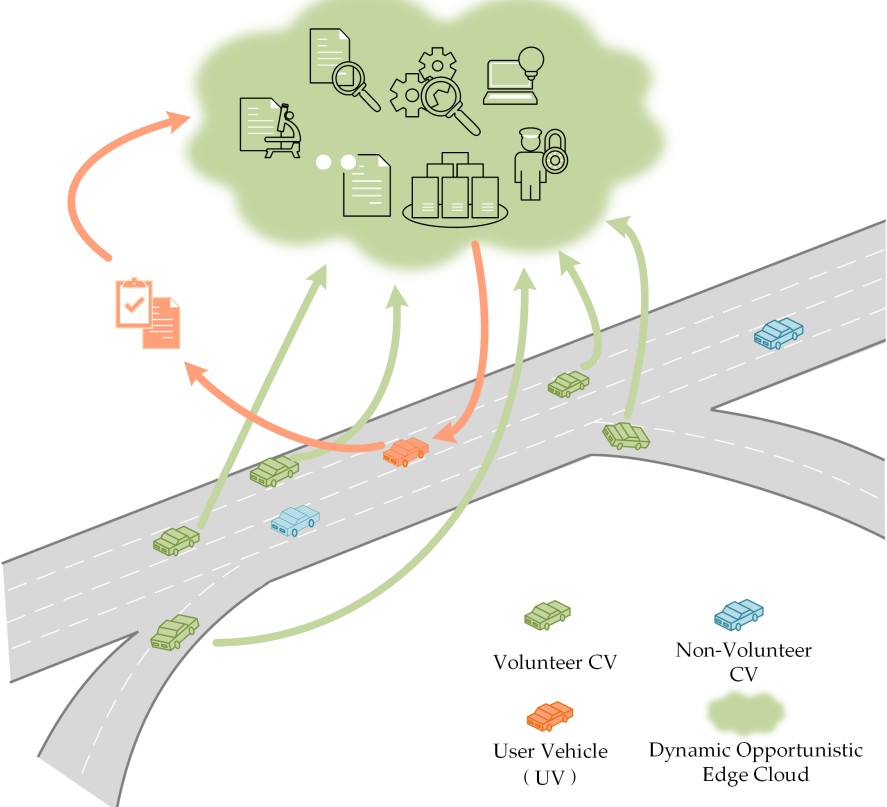

**Figure 1.** System model. Best viewed in color.

Assume that a UV with low computing power asks for a computational task offloading service from CVs with rich and surplus resources. There are C available CVs nearby, which can be represented by a set as $A = \{A_1, A_2, \ldots, A_w \ldots, A_C\}$, where $w \in [1, 2, \ldots, C - 1, C]$. However, available CVs in the vicinity should have the right to make independent decisions regarding arrival requests, which means that some of the available vehicles with rich computing resources may not be willing to share these resources with a UV. These vehicles are meaningless regarding improvements in the quality of user vehicles' computational offloading services. A binary factor $c_w$ is used to represent whether the w*th* CV is willing to provide services and be a volunteer vehicle. If w*th* CV is willing to share resources, $c_w = 1$; otherwise, $c_w = 0$. All voluntary CVs form a volunteer resource pool, which is made up of M available SVs, denoted by $N = \{N_1, N_2, \ldots, N_j, \ldots, N_M\}$, where $j \in [1, 2, \ldots, M - 1, M]$. Obviously, $N \subseteq A$ and $M \leq C$. For computing tasks, K tasks may be generated by a UV, denoted by $D = \{D_1, D_2, \ldots, D_k, \ldots, D_K\}$, in which $k \in [1, 2, \ldots, K - 1, K]$.

### 3.2. High-Speed Moving Model of CVs

When CVs drive along the expressway, there are generally no complex road conditions such as intersections or pedestrians, but the speed of CVs is usually faster than that of urban roads, and the communication links are hard to maintain. Random behaviors such as overtaking, slowing down and lane-changing also exist in the carriageway of the motorway. In addition to normal driving CVs, some vehicles may enter or leave the expressway. To

facilitate the study, consulting the model in reference [17], the movement modes of CVs in the expressway are simplified into the following three types:

1.  CV drives forward at a constant speed on the expressway;
2.  Slows down, and drives into the deceleration lane to leave the expressway;
3.  Accelerates, drives from the acceleration lane into the main lane, and joins the traffic flow.

When a CV decelerates, its speed decreases from $v_1$ to minimum speed $v_{\min}$, and then it drives at a constant speed of $v_{\min}$; the deceleration movement time of the vehicle is:

$$t_{dec} = \frac{v_1 - v_{\min}}{a_{dec}},$$

where $t_{dec}$ represents the deceleration time of a CV and $a_{dec}$ is the accelerated speed. Similarly, when a CV accelerates, its speed changes from current speed $v_0$ to $v_{\max}$, and the acceleration time is calculated as follows:

$$t_{acc} = \frac{v_{\max} - v_0}{a_{acc}},$$

$t_{acc}$ and $a_{acc}$ represent the acceleration time and accelerated velocity, respectively. Assuming that the UV travels at a constant speed of $v_{RV}$, the relative distance between SVs in the resource pool and the UV can be expressed as follows:

$$S_{rel}(t) = \begin{cases} v_{RV}t - (v_0 t + 0.5 a_{acc} t^2), accelerate, \\ v_{RV}t - (v_1 t + 0.5 a_{dec} t^2), decelerate, \\ (v_{RV} - v_j)t, uniform. \end{cases}$$

where $v_j$ is the speed of the volunteer vehicle when it runs at a constant speed. From the above formula, UV always runs at a constant speed, while the available nearby SVs may have three motion modes, namely, acceleration, deceleration or uniform motion. This article does not discuss the computation task offloading behavior of a UV when it has complex motion behavior. Only when the distance between the two workshops is within the communication range of the other party can the two vehicles communicate, thus realizing the computational task offloading. Therefore, task offloading needs to meet the following requirements:

$$|S_{rel}(t)| < 300m.$$

*3.3. V2V Communication Model*

If task k is offloaded to service vehicle (SV) j from user vehicle (UV) i, then sustainable communication links between SV j and UV i are needed to ensure a reliable task offloading process. The data rate between the two, $r_{ij}$, can be expressed as:

$$r_{ij} = W \log_2(1 + \frac{p_{ij} \times h_{ij}}{\delta^2 + I_{ij}}),$$

where i is the index of UV, j is the index of SV, W represents the bandwidth (Hz) between UV i and SV j, $p_{ij}$ is the transmission power between the two, $h_{ij}$ represents the channel gain, and $\delta^2$ and $I_{ij}$ are the noise power and interference power, respectively. Then, when task k is offloaded from UV i to SV j, the uplink transmission time is $t_{k,ij}^{up} = \frac{d_k}{r_{ij}}$. Similarly, the downlink transmission delay can be expressed as $t_{k,ij}^{down} = \frac{d_k^{back}}{r_{ij}}$, where $d_k^{back}$ denotes the data volume of results of task k. $d_k^{back}$ can be calculated as $d_k^{back} = \omega_k^g \times d_k$, in which $\omega_k^g$ represents the ratio of output to input data volume of task k. The value of g is related to the type of task k; that is, when task k is generated using critical applications, g = 1; when it is generated using high-priority applications, g = 2; when generated using low-priority applications, g = 3. $\omega_k^g$ is a positive number and is, generally, determined by the nature of

the task; its value increases with g. We explain the nature of the task in more detail in the next subsection.

Thus, the transmission time of task k is the sum of the uplink delay and downlink delay:

$$t_{k,ij}^{trans} = t_{k,ij}^{up} + t_{k,ij}^{down}.$$

### 3.4. Computing Model

In general, different tasks have different demands on diverse resources. According to the differences in the types of onboard applications, the computation requirements of tasks, and the latency requirements, the computational tasks generated by vehicles can be classified into the following three types [23,25,26]:

- Critical application (CA);
- High-priority application (HPA);
- Low-priority application (LPA).

The CA tasks generally include tasks related to driving safety and road safety, such as emergency situation responses and emergency avoidance. These tasks are directly related to the safety of drivers, passengers, pedestrians and other lives. The output data of CA tasks are usually the command type with few data. If, by any chance, such tasks are offloaded to other vehicles, not only can they not ensure the tight delay demand of computing tasks, but there is also possible package loss and offloading failure during the offloading process, which are likely to cause decision errors or timeout and thus lead to safety accidents. HPA tasks generally represent higher-priority assisted driving tasks, such as automatic parking, map navigation and some optimized safety applications. Such tasks are generally further functional expansion applications arising from the CA-type tasks to ensure the safety of drivers and passengers in order to provide a better service experience for users. The output data volume of HPA tasks is larger than that of CA-type tasks. LPA-type tasks generally include low-priority multimedia and entertainment applications, such as in-car music, movies and games, which are entertainment-type tasks with the highest tolerance for latency compared to the previous two. However, although they have a high delay tolerance, LPA tasks usually request transmissions of data of HD movies, music and online games, etc., resulting in a high ratio of output to input data. Therefore, there are grounds to assume that both HPA- and LPA-type tasks can be offloaded to other vehicles, while CA-type tasks can only be finished using local in-vehicle systems and do not involve computational offloading. We ignore CA-type tasks when discussing computational offloading and only classify HPA- and LPA-type tasks that can be offloaded. Based on the above analysis, a binary variable can be used to distinguish what type of task k is: $\beta_k = 1$ means task k belongs to HPA; otherwise, it is an LPA.

Assuming that computing task k is offloaded from UV i to SV j in the voluntary resource pool, the time cost when task k is finished, $t_{k,ij}^{exe}$, is calculated using the following:

$$t_{k,ij}^{exe} = \frac{d_k \times \rho_k}{f_j},$$

where $k \in [1, 2, \ldots, K-1, K]$ is the index of tasks, $d_k$ represents the data volume of task k in bit, $\rho_k$ denotes computational density in cycles per bit, and $f_j$ is the computing power of SV j.

Therefore, the communication and processing cost for task k when it is offloaded from UV i to SV j, $t_k^{offloading}$, can be described as follows:

$$t_k^{offloading} = t_{k,ij}^{trans} + t_{k,ij}^{exe}$$

that is,

$$t_k^{offloading} = t_{k,ij}^{up} + t_{k,ij}^{down} + t_{k,ij}^{exe}$$

then, the total task offloading delay (including communication and processing cost) for all K tasks is

$$t^{offloading} = \sum_{k=1}^{K} t_k^{offloading} = \sum_{k=1}^{K} (t_{k,ij}^{up} + t_{k,ij}^{down} + t_{k,ij}^{exe}).$$

The average completion delay (ACL) is

$$ACL = \frac{1}{K} t^{offloading} = \frac{1}{K} \sum_{k=1}^{K} t_k^{offloading} = \frac{1}{K} \sum_{k=1}^{K} (t_{k,ij}^{up} + t_{k,ij}^{down} + t_{k,ij}^{exe}).$$

*3.5. Divergent Selection Mechanism*

During the process of computational offloading, nearby available CVs actually have no obligation to provide computational services to the UV, and their own computational resources are their private property and should be freely disposed according to their subjective wishes. Correspondingly, if a resource pool of volunteer CVs has been generated, i.e., there are nearby vehicles willing to provide services for UV, the UV should also have the right to choose their offloading endpoints subjectively so as to obtain a better service experience. Considering the above two factors, we propose a divergent selection mechanism for computing the task offloading of CVs based on edge computing, taking into account both the offloading intention of the UV and the service intention of volunteer SVs, in order to satisfy the needs of both parties. We encourage nearby available CVs to contribute their computing resources and join the resource pool of volunteer SVs to provide services for user vehicles by means of reward incentives. The total system reward (TSR) for all volunteer vehicles to complete all tasks is:

$$TSR = \sum_{k=1}^{K} [\beta_k \times b_2 \times f_j \times d_k + (1 - \beta_k) \times b_3 \times f_j \times d_k],$$

where $\beta_k$ shows the type of task k; when $\beta_k = 1$, task k is an HPA-type task, and task k belongs to LPA-type if $\beta_k = 0$. $b_2$ and $b_3$ are both constants that represent the unit reward of executing HPA tasks and LPA tasks. $f_j$ is the computing power of the chosen SV j. $d_k$ is the data volume of task k. As can be seen from the above reward function, the total reward of the system is proportional to both the amount of task data and the computational power of the offloaded endpoints and is also related to the type of task and the unit reward for executing the task.

Although the above incentives are applied to available CVs in the vicinity to motivate them to contribute their computing resources, there may still be some vehicles that are unwilling to share resources due to their subjective intentions. As mentioned earlier, a binary variable $c_w$ is used to indicate whether the available vehicles are willing to perform services for the UV. The vehicle w is supposed to only be willing to contribute its computing resource when $c_w = 1$. It is also assumed that after each CV makes a decision on whether to share computing resources, the choice remains unchanged throughout the offloading process, i.e., there is no possibility for a CV to change its decision midway through computational offloading. In this way, all $c_w = 1$ vehicles form a resource pool of volunteer SVs that can provide services to the UV.

After the members of the voluntary vehicle resource pool are fixed, due to the discrepant computing power of volunteer vehicles and the unstable communication links between them, the UV has occasion to select an appropriate offloading CV to obtain better service quality. To solve this problem, a volunteer vehicle score screening mechanism is proposed to sort the existing vehicles in the volunteer pool; the computing power of the volunteer CV and the effective communication time with the UV are both comprehensively considered. The score of SV j in the voluntary resource pool can be expressed as

$$score_j = \gamma_1 \times ST_j + \gamma_2 \times CP_j,$$

where $score_j$ represents the final score of volunteer vehicle j. $ST_j$ and $CP_j$ are the stable communication times between SV j and UV i and computing power of SV j. $\gamma_1$ and $\gamma_2$ are weight coefficients, which are both positive numbers. After scoring each vehicle, all SVs in the volunteer pool are ranked according to their ratings. The vehicle with the highest rating is ranked first, obtaining an advantage over other vehicles in terms of both reliable communication time and computational capability, so priority is assigned to the highest rated vehicle when performing computational task offloading. As long as the highest ranked vehicle is still within the communication range, it will be used as the offloading endpoint. If the vehicle leaves the communication range of UV at a certain time or is still in the communication range but is about to leave and cannot complete the next computation task, the offloading endpoint is transferred to the volunteer vehicle with the second highest rating, and so on.

SVs with high ratings not only have longer communication maintenance time, but also have a relatively higher computational power. Offloading computational tasks to such SVs with priority can shorten the task computational latency and enable the UV to obtain a better service experience, while also making better use of the computational resources of resource-rich volunteer vehicles. For those volunteer SVs with lower computing power, their ratings are likely to be low, resulting in them not being a priority when performing computation offloading, which allows them to devote more of their relatively low computing power to process tasks generated by themselves (e.g., CA-type tasks). This strategy invariably balances the utilization of vehicular computational resources.

This study focuses on the computational task offloading of UVs with insufficient computational power by temporarily forming opportunistic mobile edge servers from moving CVs when roadside fixed edge servers are not available. Since not all CVs are willing to join the volunteer resource pool to contribute their computational resources, the computational capacity of the formed mobile edge server is directly related to the number of vehicles joining the volunteer pool. In order to increase the computing power of the resource pool, i.e., the mobile edge server, as many CVs as possible should be encouraged to join the volunteer pool and increase its computational capacity to complete the computing tasks of the UV within a shorter time. To achieve this goal, total computing power of the resource pool can be increased by increasing the number of volunteer vehicles, since the independent computing power of each CV is essentially fixed. The objective function can be set to maximize the total reward for all volunteer SVs in the resource pool. The higher the reward, the more likely it is that more intelligent and connected vehicles will be incentivized to join the resource pool:

$$\max\{\sum_{k=1}^{K}[\beta_k \times b_2 \times f_j \times d_k + (1 - \beta_k) \times b_3 \times f_j \times d_k]\},$$

$$st. \quad k \in [1, 2, \ldots, K-1, K],$$

$$\beta_k \in \{0, 1\},$$

$$b_2 > 0,$$

$$b_3 > 0.$$

The pseudocode of the divergent selection incentive mechanism (DSIM) proposed in this paper is shown in Algorithm 1. Based on the greedy algorithm, divergent greedy selection can be realized, maximizing the interests of both parties. In addition, DSIM can evaluate SVs' attributes in the edge cloud or resource pool, including SVs' speed, distance to the UV, computing power and QoS it can provide. Then, all SVs in the resource pool are ranked according to the evaluation results, and the top SV will be assigned priority

as the offloading destination. The maximum reward of the offloading system is finally obtained. This method can ensure that the UV obtains the best QoS and balances the resource retention of each network entity in the transportation system.

---

**Algorithm 1** DSIM Algorithm.

---

**Input:** SV set $N$; Task set $D$; User Vehicle $U$
**Output:** Maximum reward of RP, *Award*
1: **Initialize** $N$, $D$, $U$, *Flag* = $\varnothing$, *Award* = 0
2: **for** $j$ = 1,2, . . . ,$M$ in $N$ **do** // $j$ represents the index of SV
3:     $score(j) \Leftarrow \gamma_1 \times ST_j + \gamma_2 \times CP_j$ //Compute score of each SV
4:     Sort $N$ according to score // N is now sorted
5: **end for**
6: **for** $j$ = 1, 2, . . . , $M$ in $N$ **do** // N has been sorted greedily
7:     **for** $k$ = 1, 2, . . . , $K$ in $D$ **do**
8:         **if** *Flag(k)* = 1 **then** // Task $k$ has been finished
9:             **continue**
10:        **else**
11:           Distribute task $k$ to $SV_j$ // Greedy selection
12:        **end if**
13:        **if** $SV_j$ can finish task $k$ **then**
14:           *Flag(k)* $\Leftarrow 1$ //Mark that task $k$ has been finished
15:           *Award(k)* $\Leftarrow \beta_k \times b_2 \times f_j \times d_k + (1 - \beta_k) \times b_3 \times f_j \times d_k$
16:           *Award* $\Leftarrow$ *Award* +*Award(k)*
17:           **continue**
18:        **else**
19:           **break** // If $SV_j$ can not finish task $k$, skip to next SV
20:        **end if**
21:     **end for**
22:     **continue**
23: **end for**
24: **return** *Award* //Maximum reward is obtained

---

## 4. Results

In this section, we present the performance of the proposed DSIM algorithm using an Intel (R) Core (TM) i7-7500 CPU with 8 GB memory. MATLAB 2020B and Simulation of Urban Mobility (SUMO) [27,28] software are leveraged to conduct simulation experiments, in which SUMO is used to generate expressway models and simulated traffic flows, while MATLAB is used to perform our proposed algorithm. First of all, the parameter setting of simulation is introduced. Then, a series of comparative experiments are conducted to demonstrate the performance of the proposed scheme.

### 4.1. Simulation Settings

It is assumed that several CVs travel normally in a freeway segment containing one in-ramp and one out-ramp, while there are also vehicles trying to accelerate to merge or decelerate to exit the freeway. There is a UV in the traffic stream whose computing resources are insufficient that needs a nearby volunteer CV to provide auxiliary computing services. Since there is no roadside service unit or central server, the whole process is mainly achieved using V2V communication and the computational offloading incentive screening mechanism proposed in this paper. The simulation parameters are set as shown below. The communication range of each CV is 300 m. The initial velocity of each CV ranges from 40 to 100 km per hour. The user vehicle that is driving normally at a uniform speed has K = 8 computational tasks waiting to be offloaded. The values of the simulation and communication parameters are summarized in Table 1.

**Table 1.** Parameter settings.

| Parameters | Value |
|---|---|
| Bandwidth W | $2 \times 10^6$ Hz |
| Transmission Power P | 1.5 mW |
| Interference Power H | $8 \times 10^{-6}$ W |
| Noise Power N0 | $-120$ dBm |
| Communication range of CV | 300 m |
| Initial velocity of CV | 40~100 km/hour |
| Number of tasks K | 8 |
| Number of SVs M | 5~10 |
| Length of motorway segment | 10 km |

A simulation scenario was built in SUMO, where a motorway with one in-ramp and one out-ramp is modeled. As is shown in Figure 2, for the best view, the UV is colored in red, and there are vehicles in green, denoting volunteer SVs. Vehicles in blue are unwilling to share their resources and are outside of this study's scope. Note that, in order to visualize the vehicles and their intentions in the SUMO simulation, we reduced the overall size of the model to make it easier for the reader. The actual simulation length is larger than that in Figure 2, especially the part between in- and out-ramps.

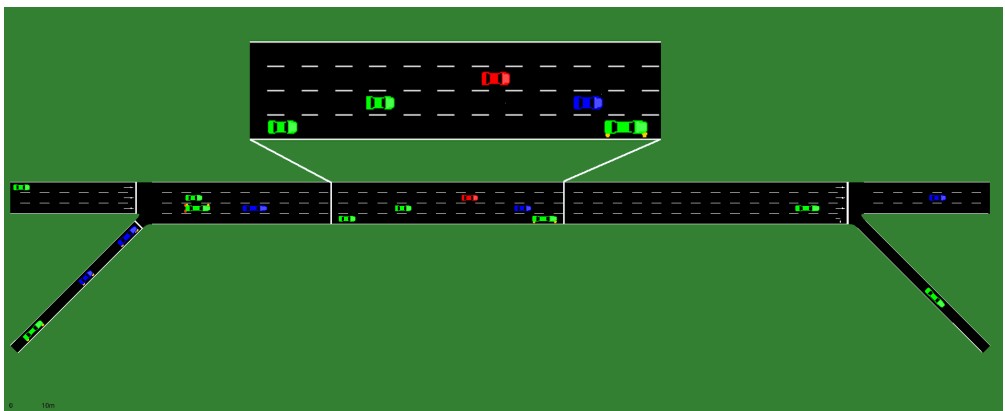

**Figure 2.** Simulation scenario in SUMO. Best viewed in color. Vehicles in red are UVs; vehicles in green are SVs (volunteer CVs); vehicles in blue are non-volunteer CVs.

*4.2. Experimental Results*

To test the feasibility of the proposed strategies, we run 1000 iterations of the Monte Carlo method, with the results averaged for statistics. Two main metrics, namely, total system reward (TSR, defined in Section 3.5.) and average completion latency (ACL, defined in Section 3.4.), are used to evaluate the efficiency of our algorithm. We show the value of TSR and ACL when average data volume varies but there are 8 fixed service vehicles (SVs), as well as when the number of SVs varies from 5 to 10. This ensures the validity of our results in terms of system performance.

Before showing the performance of our proposed solution in TSR and ACL, we first validate the superiority of our mechanism compared to others by showing the ratings of the SVs. The strategy proposed in this paper integrates the computing power, driving speed, and reliable communication time of CVs. According to the strategy, nearby intelligent CVs are first ranked and compared.

Figure 3 shows the impact of different single greedy scoring mechanisms on CV scores, in which Figure 3a shows the performance of our proposed scoring mechanisms, while Figure 3b,c,d represents the ranking of the SVs in terms of single greedy communication time, computing power and communication distance at the beginning, respectively. As can be seen, Figure 3 visualizes the basic attributes of SVs in the voluntary resource pool and is also the first step in selecting the offloading endpoint on merit. Generally speaking, the

higher the computing power, the closer the driving speed to the UV, and the longer the reliable communication time, the more desirable the SV is as a service-provider for the UV. However, the outstanding performance of any single factor is a strong not enough criterion to judge it as the optimal offloading endpoint, and only by considering various factors can we judge each nearby CV more comprehensively and make a not only more rational but more accurate decision.

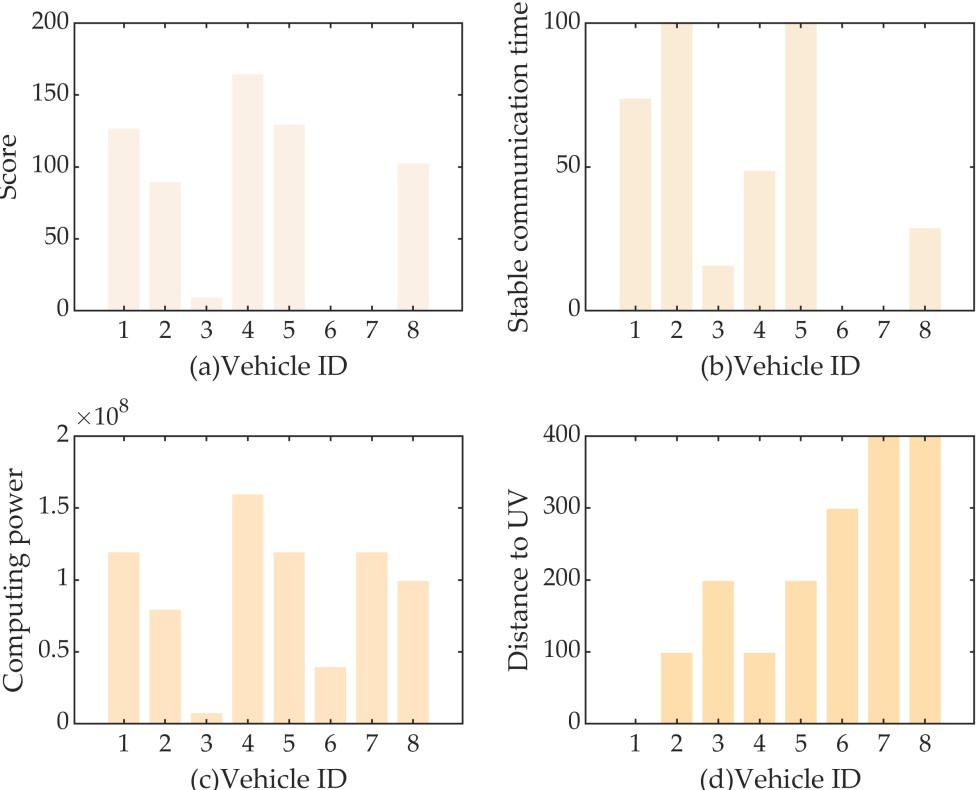

**Figure 3.** Sorting results under different scoring mechanisms. (**a**) Comprehensive score of SVs; (**b**) stable communication time in seconds for SVs; (**c**) computing power in cycles per second for SVs; (**d**) SVs' distance to UV in meters.

The effect of weight parameter changes on the total system reward is presented, as shown in Figure 4. When the weights of the CV's computing power and the weights of the reliable communication time are adjusted, the overall score of nearby CVs also changes, which changes the focus of the offloading endpoint selection strategy, leading to the final impact on the total reward value of the system. For example, when $\gamma_1 = \gamma_{11} = 0.5$, $\gamma_2 = \gamma_{12} = 2 * 10^{-7}$, the CV numbered 5 receives the highest rating and will be used as the highest-priority endpoint for computational task offloading, while when $\gamma_1 = \gamma_{21} = 0.03$, $\gamma_2 = \gamma_{22} = 3 * 10^{-7}$, the optimal unloading endpoint becomes No. 4 CV. Thus, it can be seen that the choice of parameters plays an extremely important role in decision making and the overall performance of the whole system.

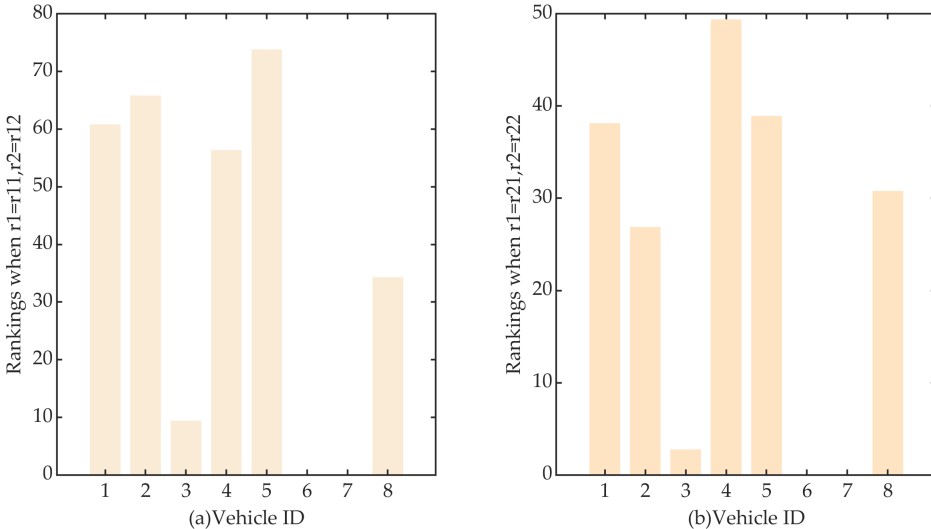

**Figure 4.** Impact of weight change on vehicle rating. (**a**) SV rankings when $\gamma_1 = \gamma_{11} = 0.5$, $\gamma_2 = \gamma_{12} = 2 * 10^{-7}$; (**b**) SV rankings when $\gamma_1 = \gamma_{21} = 0.03$, $\gamma_2 = \gamma_{22} = 3 * 10^{-7}$.

Figures 5 and 6 present the experimental results of the divergent selection method based on the incentive mechanism (DSIM) proposed in this paper with other schemes when parameters are fixed at $\gamma_1 = 0.1$, $\gamma_2 = 10^{-6}$, and the unit price of computing tasks is fixed at $b_2 = 6.5$, $b_3 = 8$, respectively. Several comparison algorithms are as follows:

- Random offloading scheme (ROS), where the UV will choose SVs for offloading randomly.
- Communication to computing ratio first (CCRF), proposed in a previous study [29]; this scheme allows the UV to choose SVs with the lowest communication to computing ratio (CCR).
- Shortest transmission time first (STTF) [30], in which the UV will choose SVs with the shortest transmission time.
- Distance-first (DF), where the UV will choose the nearest SVs to process task offloading.

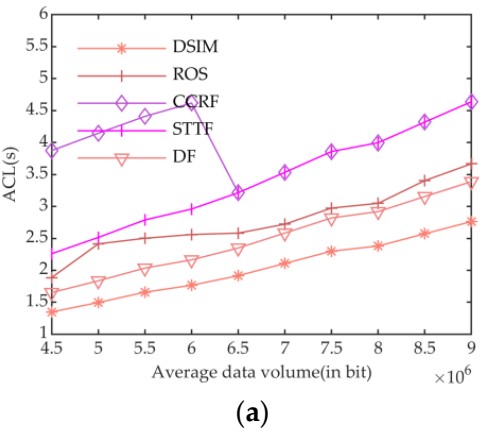

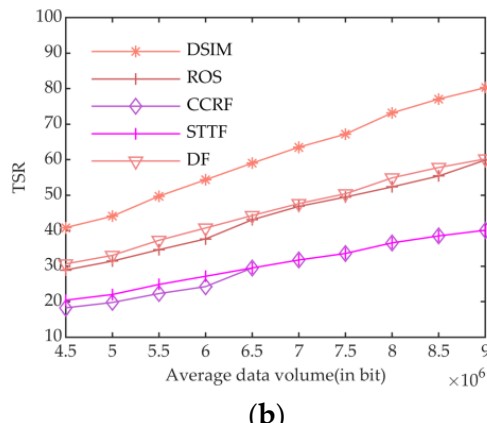

**Figure 5.** Relationship between ACL and TSR and average data volume. (**a**) Average task completion latency (ACL) versus average data volume when $\gamma_1 = 0.1$, $\gamma_2 = 10^{-6}$, $b_2 = 6.5$ and $b_3 = 8$ with 8 SVs. (**b**) Total system reward (TSR) versus average data volume when $\gamma_1 = 0.1$, $\gamma_2 = 10^{-6}$, $b_2 = 6.5$ and $b_3 = 8$ with 8 fixed SVs.

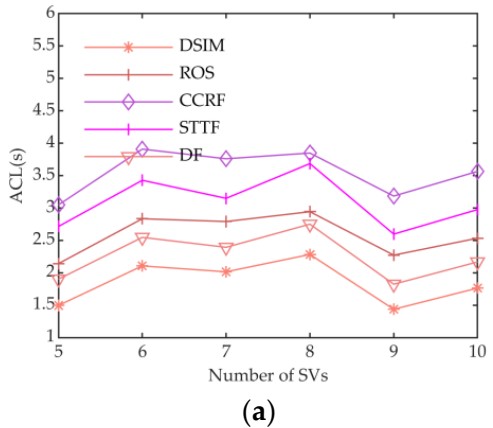
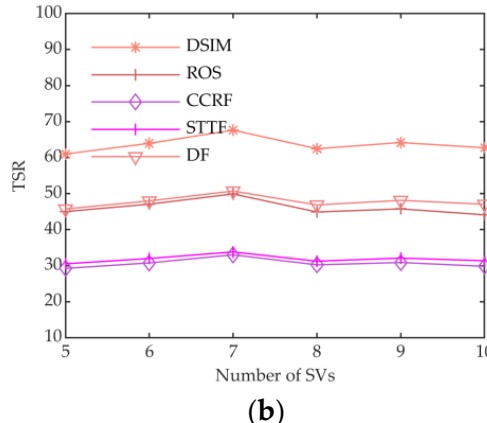

**(a)**        **(b)**

**Figure 6.** Relationship between ACL and TSR and number of SVs. (**a**) Average task completion latency (ACL) versus number of SVs when $\gamma_1 = 0.1$, $\gamma_2 = 10^{-6}$, $b_2 = 6.5$ and $b_3 = 8$ with fixed average data volume. (**b**) Total system reward (TSR) versus number of SVs when $\gamma_1 = 0.1$, $\gamma_2 = 10^{-6}$, $b_2 = 6.5$ and $b_3 = 8$ with a fixed average data volume.

It is important to note that the task type limits the determination of the task unit price. This is due to the fact that the more urgent the task is, the higher its unit price should be, so that volunteer CVs in the resource pool are more inclined to prioritize urgent tasks, meet the needs of different task characteristics, and provide efficient and reliable services to the requesting UVs. It is easy to understand that the higher the average unit price (b2 and b3) of the task, the higher the total reward received by the system. We next discuss the impact of data volume and number of SVs on ACL and TSR. Figure 5 shows how the values of ACL and TSR vary with the average data volume of the task when the number of SVs is fixed at eight. As can be seen in Figure 5a,b, the proposed DSIM always obtains the minimal average task completion delay and maximal total reward.

Figure 6 illustrates how ACL (Figure 6a) and TSR (Figure 6b) changes when the number of nearby SVs varies. It can be concluded that no matter how the SV number changes, our proposed DSIM scheme obtains the lowest average task completion delay and highest total system reward, which validates the effectiveness of our mechanism. The low latency performance of our algorithm ensures that UVs receive high-quality service, while high awards enable SVs to receive more rewards, thus satisfying the wishes of both UVs and SVs. Moreover, the rewards obtained by SVs can be used as virtual currency to offset highway tolls, parking fees, etc., which meets the demand of the intelligent transportation system.

## 5. Discussion

Based on the edge computing of connected vehicles, this paper explored computing task offloading without relying on roadside edge servers, which was realistic but lacked research. The computing tasks generated by fast-moving CVs in expressways were classified according to their characteristics, and an incentive-based divergent selection mechanism for computational offloading was proposed. The willingness of the UV and nearby available CVs was considered, and a screening mechanism was designed to score and rank the volunteer CVs. The simulation results demonstrated that the strategy we proposed outperformed other baselines considering vehicle computing power and reliable communication time, and also finely classified computation tasks to maximize the rewards for SVs in the voluntary resource pool while satisfying service quality requirements. The proposed method ensured the quality of service (QoS) of the user vehicle as well as maximizing the total reward obtained by all service vehicles. Specifically, the award obtained by service vehicles is transformed into virtual currency, which can be used to offset parking fees and highway tolls, thus providing convenience to drivers and high efficiency to the intelligent transportation system. We believe that when conflicts arise between multiple interests, the goal of balancing these interests to achieve a win–win situation is a very universal and

important idea in the field of computational task offloading of the internet of vehicles and other scientific fields.

## 6. Conclusions

The mechanism we proposed achieved good results in terms of task completion latency and overall system rewards and can respect the wishes of user vehicles and service vehicles. However, there are still some problems in this study, i.e., how to choose the appropriate weights to make the results more intuitive and make the relationship between contribution and reward more reasonable regarding its direct impact on the economic interests of drivers. In addition, we only explored single-UV, multi-SV scenarios, so it is necessary for subsequent research to focus on more complex multi-UV, multi-SV task offloading for connected vehicles. Advanced technologies such as deep reinforcement learning, digital twin, blockchain, etc., should also be considered in future research. Refining the innovation and shortcomings of this paper is of significance for follow-up studies.

**Author Contributions:** Conceptualization, methodology, software, validation, formal analysis, investigation, resources, data curation, writing—original draft preparation, visualization, S.Y.; writing—review and editing, Y.G., S.Y. and N.L.; supervision, Y.G., S.Y., N.L., D.X. and H.Y.; project administration, D.X. and H.Y.; funding acquisition, Y.G., N.L., D.X., S.Y. and H.Y. All authors have read and agreed to the published version of the manuscript.

**Funding:** This research was funded by Jiangsu Province Natural Science Fund, grant number BK20211227; National Natural Science Foundation of China, grant number 61871400, 62273356.

**Institutional Review Board Statement:** Not applicable.

**Informed Consent Statement:** Not applicable.

**Data Availability Statement:** Not applicable.

**Acknowledgments:** Not applicable.

**Conflicts of Interest:** The authors declare no conflict of interest.

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
