# Peer review of "Divergent Selection Task Offloading Strategy for Connected Vehicles Based on Incentive Mechanism"

_electronics, doi:10.3390/electronics12092143_

Round 1

Reviewer 1 Report

Divergent Selection Task Offloading Strategy for CVs Based on Incentive Mechanism >> update title as >> Divergent Selection Task Offloading Strategy for Connected Vehicles Based on Incentive Mechanism.

The abstract is confusing, rewrite it to show the background, the issue in the area of research, your proposed solution and elaborate the method you follow, and how you justify the performance of your proposed DSIM algorithm based on divergent greedy algorithm.

Remove '1.1. Background'.

Separate related work from the introduction section.

'1.2.3. Existing problems and our contribution', no need for this sub-section heading in the introduction. Write a paragraph on the rest of the paper organization at the end of the introduction section.

Write an introductory sentence for '2. Materials and Methods', before jumping to '2.1. System model'.

Write an introductory sentence for '3. Results', before jumping to '3.1 Simulation Settings'.

Give references for the literature with which you have compared your proposed as 'Random Offloading Scheme (ROS), Computing Power First (CPF), Stable Communication Time-First (SCTF), and Distance-First (DF).', but did not find it even in the related work. If these are just the variation please compare your work with on the shelf and/or recent strong related published work.

After '4. Discussion' section, write the conclusion. Reflect on the numerical findings in your concluding remarks.

enhance the reference list by including more strong related literature by query

Divergent Selection Task Offloading Strategy for... - Google Scholar

some examples as

Deep Reinforcement Learning for Offloading and Resource Allocation in Vehicle Edge Computing and Networks - https://doi.org/10.1109/TVT.2019.2935450

Task offloading paradigm in mobile edge computing-current issues, adopted approaches, and future directions - https://doi.org/10.1016/j.jnca.2022.103568

Priority-Aware Task Offloading in Vehicular Fog Computing Based on Deep Reinforcement Learning - https://doi.org/10.1109/TVT.2020.3041929

Many-to-Many Task Offloading in Vehicular Fog Computing: A Multi-Agent Deep Reinforcement Learning Approach - https://doi.org/10.1109/TMC.2023.3250495

Data Security Through Zero-Knowledge Proof and Statistical Fingerprinting in Vehicle-to-Healthcare Everything (V2HX) Communications - https://doi.org/10.1109/TITS.2021.3066487

RL/DRL Meets Vehicular Task Offloading Using Edge and Vehicular Cloudlet: A Survey - https://doi.org/10.1109/JIOT.2022.3155667

Reviewer 2 Report

The paper discusses using edge computing in Intelligent Transportation Systems (ITS) for high-speed Connected Vehicles (CVs) without the aid of roadside edge servers. The authors propose a Divergent Selection task offloading strategy based on an Incentive Mechanism (DSIM) to balance the interests of users and service vehicles. The DSIM algorithm uses a divergent greedy algorithm to solve the problem, and simulation results show its effectiveness.

The text is well written, justifying, in a general way, the choices made and providing the necessary descriptions for a good understanding of the solution, the experiments performed, and the results obtained. 

Some errors in agreement and typing are found in the text. 

The article follows the format provided. 

The mathematical formulation and algorithm appear correct.

 It would be appropriate to provide a better comparison with the literature in section 1.2. Related work, highlights the advantages and differences of this approach compared to existing ones.

Related to the affirmation in lines (213-214) answer ” When an available CV becomes a member of the resource pool, it will be rewarded according to its available communication time, computing power, and service delivery timeWhat is the duration of this operation, is it fast enough to be completed before the VANET topology changes?

In Section 2, it is recommended to provide clear definitions for all variables used in the equations, such as the variable w n j.

The author does not point out the implementation costs for the proposed solution.

In lines 305-309 “When the computational task is finished, similar to previous studies [10][19], we ignore the downlink delay of computation result return due to the relatively small amount of data of the result. Thus, the communication delay between the two vehicles where the computation offloading occurs can be approximated as the uplink transmission delay of the computation task” 

The statement is not appropriate because due to the movement of vehicles and processing time, the network may be significantly altered. Additionally, it is possible that the two vehicles intended to communicate may have lost connection due to distance or network interference.

The proposed solution disregards the direction of the vehicles, which can have a significant impact on the solution's real-world effectiveness and accuracy.

Regarding the Results section:

How was the network and traffic simulation performed?

The simulation should have used a more appropriate and realistic framework for analyzing the solution, such as Veins, Sumo, and Omnet. These frameworks would significantly reduce the gap between reality and simulation.

The results should be compared with other solutions in the literature.

In the Parameter settings, it shows that only 8 SVs were used. It would be desirable to vary this value to understand the behavior of the solution.

What is the length of the simulated highway section?

How many entrances and exits are there? A more detailed description of the highway is necessary.

What is the communication and processing cost of this solution?

What statistics were used to ensure the results?

Good luck with improving the article!

Reviewer 3 Report

This paper explores the application of edge computing in the construction of Intelligent Transportation Systems (ITS), specifically in the context of the Internet of Vehicles (IoV). The authors investigate the offloading of computational tasks in vehicular edge computing for high-speed Connected Vehicles (CVs), without the assistance of roadside edge servers. In this approach, user vehicles with insufficient computing power offload some of their tasks to nearby CVs with more abundant resources. The authors develop a task classification model for CVs and explore a high-speed driving model to refine the task offloading process.

Drawing on Game Theory, the authors design a Divergent Selection task offloading strategy based on Incentive Mechanism (DSIM) that balances the interests of both user and service vehicles. The DSIM algorithm allows user vehicles to select offloading endpoints based on merit to optimize Quality of Service (QoS), while service vehicles are given the autonomy to contribute their resources or not, as opposed to being obligated to do so by default. CVs that contribute resources are rewarded to incentivize more CVs to participate. The authors introduce the DSIM algorithm based on a divergent greedy algorithm to solve the proposed problem and experimental simulations validate the effectiveness of their mechanism.

Overall, this paper provides a valuable contribution to the understanding of computational task offloading in vehicular edge computing for high-speed Connected Vehicles, without relying on roadside edge servers. The proposed Divergent Selection task offloading strategy based on Incentive Mechanism is shown to be effective in balancing the interests of both user and service vehicles and optimizing QoS.

The authors must improve the following items in the final version of the manuscript.

1.     I noticed that authors started a section and immediately started a subsection in it without any text.

2.     The authors should clarify the simulation tool they used and specify the goals of their simulations, including which parameters are fixed and which are being varied.

3.     There are no conclusions in the paper. Additionally, the authors need to include limitations of the study and future research.

Round 2

Reviewer 1 Report

All comments are very well addressed.